# LANGUAGE IDENTIFICATION IN THE LIMIT WITH COMPUTATIONAL TRACE

**Binghui Peng**
University of Maryland, College Park
binghuip@umd.edu

**Amin Saberi**
Stanford University
saberi@stanford.edu

**Grigoris Velegkas**
Google Research
gvelegkas@google.com

## ABSTRACT

Training on Chain-of-Thought (CoT) traces has been empirically shown to dramatically improve the capabilities of Large Language Models (LLMs), yet a formal understanding of its power remains limited. In this work, we investigate the role of training on such computational traces from the perspective of language learnability. We introduce a new learning model, *identification in the limit with trace*, which augments Gold's classic paradigm (Gold, 1967) by providing the learner not only with examples from a target language but also with computational traces from the machine that accepts them.

Our results reveal that access to these traces dramatically enhances the power of the learner. We first prove that with perfect computational traces, the class of all recursively enumerable languages (those recognizable by Turing Machines) becomes identifiable in the limit. This stands in sharp contrast to Gold's famous impossibility result, which holds even for the simple class of languages that are recognizable by deterministic finite automata. We then analyze the more challenging scenario where the learner has only partial information regarding the computational traces, which are also subject to adversarial corruptions. In this setting, we establish a set of trichotomic results on the amount of error that can be tolerated for the successful identification of language classes across the Chomsky hierarchy.

## 1 INTRODUCTION

Large Language Models (LLMs) have demonstrated remarkable capabilities across a vast array of complex tasks, yet they can still falter on seemingly elementary problems. A fascinating empirical discovery is that their performance on such tasks often improves dramatically when they are trained on a corpus of data that contain not only input-output pairs, but also a "chain-of-thought" (CoT), *i.e.*, a step-by-step explanation of the reasoning process (Wei et al., 2022; Chung et al., 2024). Despite these important practical successes, a comprehensive theoretical understanding of the benefits of reasoning traces in the training of LLMs remains elusive.

Prior work has begun to formalize the usefulness of different types of CoT traces in the training process through theoretical models that are primarily statistical in nature (Malach, 2023; Joshi et al., 2022; Altabaa et al., 2025). In our work, we adopt a perspective that is complementary to these statistical approaches and seeks to understand if CoT training helps LLMs learn the correct *world model* (Vafa et al., 2024), *i.e.*, the correct "rules" that describe the environment they operate in. In order to make progress on this question, we need to first give a mathematical definition of the world model. For that, we rely on a classical work from Gold (Gold, 1967) who introduced the notion of *language identification in the limit*. In Gold's model there is a countable collection of strings $\mathcal{X}$ (*e.g.*, the set of all finite-length binary strings) and a countable collection of languages $\mathcal{L}$, where each language $L$ is defined as a (potentially infinite) subset of $\mathcal{X}$. Every language can be thought of as representing all the "facts" within some world. For instance, building on the ideas of

Li et al. (2023); Nanda et al. (2023) who used the task of learning rules of board games to test if LLMs are capable of learning world models, we can think of different languages in $\mathcal{L}$ as representing the valid states of different board games. Gold's model formalizes learning as a two-player game where an adversary chooses some $K \in \mathcal{L}$ and presents an *enumeration* of it to the learner, who must eventually converge to a correct description of that language, after making a *finite* number of mistakes. It is well-established that under this paradigm, most of the interesting classes of languages are *not* identifiable (Angluin, 1980). Thus, it is natural to ask: if we augment the Gold–Angluin model with a notion of chain-of-thought, can we expand the class of languages that are identifiable in the limit? We introduce a new model, *identification in the limit with traces*, where the learning algorithm observes not only examples from the target language but also a computational trace, which in our model is defined as the sequence of computational steps (*i.e.*, "trace") that a fixed machine takes to accept them.

Our first result shows that, perhaps surprisingly, the class of all languages accepted by Turing Machines (TMs) is identifiable with traces. This is in striking contrast to the negative results in Gold's setting, who showed that the much simpler class of regular languages, *i.e.*, those that are recognizable by DFAs, is *not* identifiable in the limit (Gold, 1967).

**Informal Theorem 1** (Informal statement of Theorem 3.1). *The class of languages recognized by Turing machines is identifiable in the limit with traces.*

Having established these strong positive results, we turn to a crucial question of robustness: how resilient are these learning guarantees to imperfections in the provided traces? To model these imperfections, we consider scenarios where the computational traces are corrupted—this includes traces that are only partially observable, skip important steps, or contain errors. As a unified metric, we measure this corruption as the deviation from the true computational trace, quantified by the edit distance. In this setting, we establish tight bounds on the amount of error a learner can tolerate while still successfully identifying regular languages (recognizable by DFAs), deterministic context-free languages (recognizable by Deterministic Pushdown Automata, DPDAs), and recursively enumerable languages (recognizable by TMs).

**Informal Theorem 2** (Informal statement of Theorem 4.1 4.3 4.5). *Consider the task of language identification in the limit with traces:*

- *For languages recognizable by DFAs, identification is possible even with a constant fraction of corruption (for any constant rate less than 1).*

- *For languages recognizable by DPDAs, identification is impossible for any constant fraction of corruption, but becomes possible when the corruption rate is diminishing.*

- *For languages recognizable by TMs, identification is possible if and only if the total number of errors per trace is bounded.*

Formal definitions of our corruption model are provided in Definitions 4 and 5. Informal Theorem 2 reveals a varied landscape of identifiability corresponding to the error rate and the language's complexity. For the most powerful model (TMs), identification can only be achieved if the number of errors present in an example is finite. In contrast, for the weakest model (DFAs), identification remains possible even when a 0.999 fraction of the computational trace is corrupted (per example). The intermediate model (DPDAs) occupies a middle ground: it cannot tolerate even a 0.01 fraction of error, but identification becomes possible when the corruption rate diminishes with the length of the computational trace.

## 1.1 RELATED WORK

**Identification and Generation in the Limit.** The paradigm of language identification in the limit was introduced in the pioneering work of Gold (1967) and learnability in this setting was fully characterized by Angluin (1979; 1980). Gold's setting has been very influential both in learning theory and in computational linguistics; we refer the interested reader to Lange et al. (2008) for a comprehensive survey of results in this area. Inspired by this line of work, Kleinberg & Mullainathan (2024) recently introduced the notion of *generation* in the limit: instead of having to exactly identify the target language, like in Gold's setting, the learner here is required to eventually start producing *valid unseen* elements of the language, *i.e.*, elements of the target language that have not appeared in

the input it has witnessed so far. The finding of Kleinberg & Mullainathan (2024) is rather surprising: under this modified objective *all* countable collections of languages become generatable in the limit. This result has led to a flurry of follow-up work. Li et al. (2024) further developed this notion using learning-theoretic tools and studied a hierarchy of notions of generation. Moreover, several works have formalized and studied trade-offs between *breadth* and *hallucinations* in language generation (Kalavasis et al., 2024b;a; Charikar & Pabbaraju, 2024; Peale et al., 2025; Kleinberg & Wei, 2025a;b). Raman & Raman (2025); Bai et al. (2025); Mehrotra et al. (2025); Li & Zhang (2026) studied notions of *noisy* language generation, while Hanneke et al. (2025); Bai et al. (2025) showed that generation is *not* closed under finite unions.[1] Inspired by these lines of work, Karbasi et al. (2025) formalized and studied a setting of *automated hallucination detection*. Recently, Li et al. (2026) formalized a notion of generation in continuous spaces, and Høgsgaard & Pabbaraju (2026) studied a notion of agnostic identification and generation. Lastly, Anastasopoulos et al. (2026) proposed a setting of *safe* language generation in the limit.

**Learning Algorithms in the Limit.** The most closely related setting to ours is the very recent work of Papazov & Flammarion (2025). They consider a different version of the identification problem where the goal is to learn *recursive functions* in the limit, and the learner receives as input a stream of evaluations of the target function on *every* point of the domain. In the context of binary functions, this means that the learner observes both positive and negative examples, of the target language, whereas in our setting it receives only positive examples. This difference is subtle yet critical; in the presence of negative examples, identification in the limit becomes trivial as one can simply output the smallest-indexed language that is consistent with the enumeration so far. Nevertheless, there exist classes of languages such that the above task is *undecidable*. The work of Papazov & Flammarion (2025) focuses on the computability of the task and proves that with certain side information (such as the computation time of the target machines), the identification task also becomes decidable. These results are derived with the presence of negative examples so they are fundamentally different from our work.

**Theory of CoT.** Recent theoretical work has advanced our understanding of chain-of-thought (CoT) reasoning by grounding it in formal statistical learning frameworks. Malach (2023) studies autoregressive next-token prediction in a PAC-style setting, showing that even linear models trained on i.i.d. examples can approximate any Turing-computable function when allowed to emit sufficiently long intermediate sequences. This introduces a new statistical complexity measure—length complexity—which captures the tradeoff between reasoning depth and learnability. Joshi et al. (2022) extend this perspective by formalizing a time-invariant autoregressive learning setup, where reasoning steps are repeated applications of a shared predictor. Their analysis provides generalization bounds and sample complexity guarantees for learning with and without intermediate supervision, showing that sample complexity can remain independent of the CoT length under certain conditions. Altabaa et al. (2025) adopt an information-theoretic approach, quantifying the statistical advantage of CoT supervision via a new measure called CoT information. Their results demonstrate that observing intermediate reasoning steps can provably reduce sample complexity, with matching upper and lower bounds. Since these works study statistical settings, they are orthogonal to the focus of our paper. On a related note, a recent line of work (Liu & Moitra, 2025; Gaitonde et al., 2025; Wen et al., 2025) also shows that, with the presence of computation trace, one could circumvent computational lower bounds in various statistical learning settings.

## 2 MODEL

We begin by recalling the classic paradigm of language learnability, followed by our main definition which augments this model with computational traces.

**Definition 1** (Identification in the Limit (Gold, 1967)). *Let $\mathcal{L}$ be a countable class of languages over a finite alphabet $\Sigma$. An infinite sequence of strings $E = (x_1, x_2, \ldots)$ is an **enumeration** of a language $K \in \mathcal{L}$ if the set $\{x_t\}_{t \in \mathbb{N}}$ is equal to $K$. The class $\mathcal{L}$ is **identifiable in the limit** if there exists an algorithm $I$ that, for any $K \in \mathcal{L}$ and any enumeration $E$ of $K$, converges to a correct representation of $K$. That is, there exists a time $t^*$ such that for all $t \geq t^*$, the algorithm's guess $I_t(x_1, \ldots, x_t)$ is constant and correct.*

---

[1] This result holds for *uncountable* collections of languages, which are outside the scope of our work.

Below, we give a concrete instantiation of this definition inspired by Vafa et al. (2024) who evaluated LLMs on the task of reconstructing maps of cities.

**Example 1.** *We instantiate Definition 1 by interpreting languages as city maps, where each language represents all valid routes within a city. Let $\Sigma$ be a (finite) set of vertices representing intersections, and $\mathcal{X}$ the set of all finite sequences of elements of $\Sigma$. We let $\mathcal{G}$ be the set of all graphs over $\Sigma$, where each graph $G = (V, E) \in \mathcal{G}$ models a city layout, with edges representing roads. The class of languages is then $\mathcal{L} = \{L_G \mid G \in \mathcal{G}\}$. A specific language $L_G$ consists of all valid routes (paths) in the graph $G$; formally, a string $w = \sigma_1 \sigma_2 \ldots \sigma_k \in \mathcal{X}$ is in $L_G$ if and only if $(\sigma_j, \sigma_{j+1}) \in E$ for all $1 \le j < k$. Finally, for a target city $K \in \mathcal{L}$, an enumeration $E = (r_1, r_2, \ldots)$ is an infinite sequence of observed valid routes such that the set $\{r_t\}_{t \in \mathbb{N}}$ equals $K$.*

We consider a set of canonical models of computation: Deterministic Finite Automata, Deterministic Pushdown Automata, and Turing Machines. These are standard textbook models of computation (Sipser, 1996) and they capture an increasing complexity of the underlying languages (regular language, deterministic context free grammar and recursively enumerable languages) and the corresponding identification tasks. We present a brief introduction to these models in Definition 2; for a comprehensive overview, we refer to the textbook of Sipser (1996). In the following definition, each machine is formalized as a tuple comprising several core components. $Q$ denotes a finite set of states representing the machine's possible internal configurations. $\Sigma$ is the input alphabet, *i.e.*, a finite set of symbols that form the input strings. Computation begins at a designated start state, $q_0 \in Q$. A subset of states, $F \subseteq Q$, is defined as the set of final (or accepting) states. The behavior of the machine is governed by a transition function, $\delta$, which dictates how the machine changes state and manages its memory in response to the input it reads. The distinctions of DFAs/DPDAs/TMs lie in their specific memory structures and the definition of $\delta$. To implement memory, models utilize an additional alphabet $\Gamma$ (representing the stack or tape contents). They may also employ specific constants such as $Z_0$ for the initial stack symbol, $B$ for the blank tape symbol, and directions $\{L, R\}$ to indicate the movement of the tape head.

**Definition 2** (Computational Models). *A **Deterministic Finite Automaton (DFA)** is a 5-tuple $(Q, \Sigma, \delta, q_0, F)$ with a transition function $\delta : Q \times \Sigma \to Q$. Here $Q$ denotes the set of states, $\Sigma$ is the set of input symbols, $q_0$ is the initial state, and $F \subseteq Q$ are the accepting (final) states.*

*A **Deterministic Pushdown Automaton (DPDA)** is a 7-tuple $(Q, \Sigma, \Gamma, \delta, q_0, Z_0, F)$ that includes a set of states $Q$, an input alphabet $\Sigma$, a stack alphabet $\Gamma$, the initial state $q_0$, an initial stack symbol $Z_0 \in \Gamma$, a set of accepting states $F \subseteq Q$ and a transition function $\delta : Q \times (\Sigma \cup \{\epsilon\}) \times \Gamma \to Q \times \Gamma^*$.*

*A **Turing Machine (TM)** is a 7-tuple $(Q, \Sigma, \Gamma, \delta, q_0, B, F)$ that includes a set of states $Q$, an input alphabet $\Sigma$, a tape alphabet $\Gamma \supseteq \Sigma$, the initial state $q_0$, a blank symbol $B \in \Gamma \setminus \Sigma$, a set of accepting states $F \subseteq Q$ and a transition function $\delta : Q \times \Gamma \to Q \times \Gamma \times \{L, R\}$.*

We now introduce our primary object of study, which equips the learner with step-by-step computational traces (like CoT).

**Definition 3** (Identification in the Limit with Traces). *Let $\mathcal{L}$ be a class of languages and $\mathcal{M}$ be a class of machines. For a machine $M \in \mathcal{M}$ and an input $x \in L(M)$, let $c_M(x)$ be the accepting sequence of computational steps of $M$ on $x$. An **enumeration of $K$ with traces from $M$** is a sequence $E_{trace} = ((x_1, c_M(x_1)), (x_2, c_M(x_2)), \ldots)$ where $(x_t)$ is an enumeration of $K = L(M)$. The class $\mathcal{L}$ is **identifiable in the limit with traces from $\mathcal{M}$** if there exists an algorithm $I$ that converges to a correct representation of $K$ when given any such $E_{trace}$.*

When the underlying machine and the input $x$ are clear from context, we write $c$ for $c_M(x)$. Let us briefly elaborate on the notion of trace we consider in our work. At every timestep of its execution, every computational machine we are considering can be described by a set of parameters (*e.g.*, its current state, memory tape, position on the tape etc.). A computational trace simply describes the set of (changing) parameters of the underlying machine in every timestep. Thus, the length of the trace is the number of computational steps the machine needs to accept the given string. For a concrete example in the case of DFAs we refer the reader to Example 2.

**Example 2** (DFA Computational Trace). *To illustrate the concept of a computational trace (CoT) as defined in Definition 3, we provide a concrete example using a Deterministic Finite Automaton (DFA). For a DFA, the machine's configuration at any point in time is entirely determined by its current state. Thus, the computational trace is simply the sequence of states visited. Let us define a DFA $M = (Q, \Sigma, \delta, q_0, F)$ over the alphabet $\Sigma = \{0, 1\}$ that recognizes the language $L$*

*consisting of all strings containing an **even** number of 1s. Here, $Q = \{q_{even}, q_{odd}\}$, the starting state is $q_0 = q_{even}$, and the accepting state is $F = \{q_{even}\}$. The transition function $\delta$ is defined as $\delta(q_{even}, 0) = q_{even}, \delta(q_{odd}, 0) = q_{odd}, \delta(q_{even}, 1) = q_{odd}, \delta(q_{odd}, 1) = q_{even}$. Pictorially, the DFA is shown in Figure 1. Suppose the learner is presented with the input string $x = 1010$. The execution*

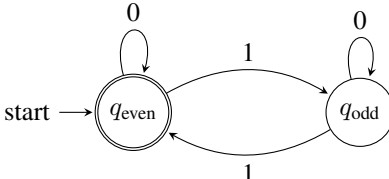

Figure 1: DFA $M$ accepting strings with an even number of 1s.

*of $M$ on $x$ proceeds as follows: start in state $\boldsymbol{q_{even}}$, transition to $\boldsymbol{q_{odd}}$, stay in $\boldsymbol{q_{odd}}$, transition to $\boldsymbol{q_{even}}$, and, finally, stay in $\boldsymbol{q_{even}}$. Thus, the computational trace $c_M(x)$ is the complete sequence of states visited during this execution: $c_M(1010) = (q_{even}, q_{odd}, q_{odd}, q_{even}, q_{even})$.*

Lastly, we provide the notion of *robust* identification in the limit with CoT, where we allow for imperfect traces that differ from the correct one in edit distance.

**Definition 4** (Robust Identification in the Limit with Traces). *The task of robust identification has a similar setup as language identification with traces, except that now the $\widetilde{c}_M(x_t)$ the algorithm observes is noisy, i.e., it is a corrupted version of the true trace $c_M(x_t)$ with edit distance error $e_M(x_t)$.[2] The criterion for identification is the same as in Definition 3, where the true trace is replaced with the corrupted one.*

**Assumption 1** (States in the Corrupted Trace). *We assume that $\widetilde{c}_t$ (the corrupted trace) does not introduce non-existent states of the machine, i.e., if $Q$ is the set of states of the machine, the corrupted trace does not introduce states that are not in $Q$.*

To give a concrete example of the notion of error we allow for, consider the DFA in Example 2 where the string is $x = 1010$ and the true trace is $(q_{even}, q_{odd}, q_{odd}, q_{even}, q_{even})$. Corrupted traces for that string can be (among others)

- $(q_{even}, q_{even}, q_{odd}, q_{even}, q_{even})$, which has edit distance 1 to the original (substitute the second element);

- $(q_{even}, q_{odd}, q_{even}, q_{even})$, which has edit distance 1 to the original (delete the third element);

- $(q_{even}, q_{odd}, q_{odd}, q_{odd}, q_{odd}, q_{even}, q_{even})$, which has edit distance 2 to the original (insert two $q_{odd}$'s).

It is clear that whether language identification can be done depends on the edit distance error. On the one hand, if the error is always zero, then it is equivalent to the language identification with traces (Definition 3). On the other hand, if the edit distance error is equal to or greater than the length of the actual trace, the trace presents no useful information and it goes back to the original language identification task (Definition 1). Our goal is to understand the intermediate settings between these two extremes, which we divide into three regimes:

**Definition 5** (Error regime). *We consider the following three error regimes:*

- *Constant error rate: There is a fixed constant $\alpha \in (0, 1)$, such that the edit distance error rate is bounded by $\alpha$ asymptotically. Formally, there exists a pair of constants $\alpha, \ell_\alpha$ such that for any input string $x$ with length at least $\ell_\alpha$ ($|x| \geq \ell_\alpha$), its corruption error is at most an $\alpha$-fraction of actual computation trace, i.e., $e_M(x)/|c_M(x)| \leq \alpha$.*

- *Diminishing error rate: The edit distance error rate is $o(1)$ asymptotically. Formally, for any constant $\alpha \in (0, 1)$ there exists a constant $\ell_\alpha$ (depending on the value of $\alpha$) such that for any input string $x$ of length at least $\ell_\alpha$ ($|x| \geq \ell_\alpha$), the corruption error is at most an $\alpha$-fraction of the actual computation trace, i.e., $e_M(x)/|c_M(x)| \leq \alpha$.*

---

[2]Edit distance is the minimum number of single-character insertions, deletions, or substitutions required to transform one string into another.

- *Finite error: The edit distance is bounded by an absolute constant $C$ asymptotically. Formally, there exists a constant length $\ell_C$ such that for any input string $x$ of length at least $\ell_C$ ($|x| \geq \ell_C$), the edit distance error is bounded by $C$, i.e., $e_M(x) \leq C$.*

For all the results of this paper, we assume the algorithm knows which error regime it operates in. Moreover, for the constant error rate regime, the constant $\alpha$ and $\ell_\alpha$ are known in advance; for the diminishing error rate regime, $(\alpha, \ell_\alpha)$ are known for all $\alpha \in (0,1)$, while for the finite error, the algorithm knows the constant $C$ and $\ell_C$.

## 2.1 USEFUL FACTS

Below we state some facts from prior work that are useful for our derivations.

**Fact 2.1** ((Gold, 1967)). *Language identification (without CoT) can be done when the class of languages $\mathcal{L}$ is finite.*

*Proof Sketch.* At any time step $t$, the identification algorithm outputs the index $\widehat{i}_t$ of a language in $\mathcal{L}$ that is *consistent* with the input $x_1, \ldots, x_t$ and is *minimal* with respect to the subset ordering. To see why this leads to identification in the limit, first notice that after some finite timestep $t^*$ all the languages that are consistent with the input are the target language $K$ and its supersets (this is because $\mathcal{L}$ is finite, so any language that is not a superset of $K$ will eventually be contradicted by the input). Thus, outputting a minimal language after $t^*$ leads to correct identification. $\square$

The next result illustrates the difficulty of identification of countable collections in the absence of CoT. We describe a simple collection of *regular* languages that is not identifiable in the limit. The crux of the difficulty is that the algorithm never receives information about strings that are *not* in the target $K$, *i.e.*, it receives only positive examples.

**Fact 2.2** ((Gold, 1967)). *Language identification (without CoT) is impossible for the language collection $\mathcal{L} = \{\mathbb{N}, L_1, L_2, \ldots\}$, where $L_i = \{1, 2, \ldots, i\}$.*

*Proof Sketch.* Assume towards contradiction that there exists some algorithm that identifies $\mathcal{L}$ in the limit. Consider the adversary that starts enumerating 1 for multiple timesteps. Since the algorithm identifies in the limit, there must be some timestep $t_1$ that it identifies $L_1$ (otherwise the algorithm fails since the adversary can keep enumerating 1 indefinitely). Now at timestep $t_1 + 1$ the adversary enumerates 2 and keeps doing so until the algorithm identifies $L_2$. The construction continues in this fashion. Hence, the adversary ends up enumerating $\mathbb{N}$ and it constructs an infinite sequence of timesteps $t_1, t_2, \ldots$, in which the algorithm makes a mistake. $\square$

## 3 IDENTIFICATION WITH PERFECT TRACE

We start with the result of identification in the limit with traces that does not contain any noise.

**Theorem 3.1.** *Let $\mathcal{M}$ be the collection of all TMs and $\mathcal{L} = \{L_1, L_2, \ldots\}$ where every $L \in \mathcal{L}$ is recognized by some $M \in \mathcal{M}$. Then, $\mathcal{L}$ is identifiable in the limit with trace from $\mathcal{M}$.*

As a warm-up, it is instructive to prove the result for the simpler setting where $\mathcal{M}$ is the set of all DFAs. It is worth highlighting that without access to computational trace, *i.e.*, in the original model of Gold, this collection is *not* identifiable in the limit (see Fact 2.2). Due to space constraints, we present the proof of the theorem for Turing machines in Appendix A.

**Identification of DFAs (sketch).** Fix a binary alphabet $\Sigma = \{0, 1\}$. Let $M^* = (Q^*, \Sigma, \delta^*, q_0, F^*)$ be a DFA for the unknown target regular language $K$. For $x \in K$, let $c(x)$ be the (unique) accepting state sequence visited by $M^*$ on $x$, including the final accepting state.

**Learner.** At every step $t$ maintain a DFA $M_t = (Q_t, \Sigma, \delta_t, q_0, F_t)$ with a reject sink $r \notin F_t$.

1. Initialization: $Q_0 = \{q_0, r\}$, $F_0 = \emptyset$, and $\delta_0(q, a) = r$ for all $q \in Q_0$, $a \in \{0, 1\}$.

2. Upon receiving $(x_t, c(x_t))$, let $\widetilde{Q}_t$ be the set of states occurring in $c(x_t)$ and set $Q_t \leftarrow Q_{t-1} \cup \widetilde{Q}_t$. For every *newly observed* state $q \in \widetilde{Q}_t \setminus Q_{t-1}$ and $a \in \{0, 1\}$, set provisionally $\delta_t(q, a) \leftarrow r$. If the last state of $c(x_t)$ is $q^{\text{acc}}$, then set $F_t \leftarrow F_{t-1} \cup \{q^{\text{acc}}\}$.

3. Traverse $c(x_t), x_t$ and for each observed transition $(q, a, q')$, set $\delta_t(q, a) \leftarrow q'$.

**Invariant (subset property).** For all $t$, the language recognized by the constructed DFA satisfies $L(M_t) \subseteq K$. By construction, $M_t$ differs from $M^*$ in two conservative ways: (i) it may *omit* states that have not yet appeared on any accepting trace (so runs that would visit those states in $M^*$ are simply unavailable in $M_t$ and lead to rejection), and (ii) for any $q \in Q_t$ and $a \in \Sigma$ whose outgoing edge has not been witnessed, it maps $(q, a)$ to the non-accepting sink $r$; on witnessed pairs it copies $\delta^*$ exactly. Hence every accepting run of $M_t$ from $q_0$ stays within $Q_t$ and uses only witnessed edges, which coincide with the corresponding edges of $M^*$. Moreover, the run can end only in a state in $F_t \subseteq F^*$ (accepting states are added only when witnessed). Therefore the same input is accepted by $M^*$, and $L(M_t) \subseteq L(M^*) = K$.

**Eventual completeness.** Let $Q^+ \subseteq Q^*$ and $E^+ \subseteq Q^* \times \Sigma$ be the states and labeled edges that lie on at least one accepting computation of $M^*$. For every $q \in Q^+$ there exists some $x \in K$ whose accepting trace visits $q$; hence each $q \in Q^+$ appears in some $c(x)$ and is added to $Q_t$ after finitely many steps. Likewise, for each $(q, a) \in E^+$ there exists some $x \in K$ whose accepting trace uses that transition, so $\delta_t(q, a)$ is eventually overwritten to $\delta^*(q, a)$. Since $E^+$ is finite, there is a finite time $t^\star$ after which $\delta_t(q, a) = \delta^*(q, a)$ for all $(q, a) \in E^+$ and $Q_t$ contains all of $Q^+$. Thus, after some time $t^\star$, every accepting path of $M^*$ is also an accepting path of $M_t$, and by the subset invariant, $M_t$ accepts no string outside $K$. Hence $L(M_t) = K$ for all $t \geq t^\star$.

## 4 ROBUST IDENTIFICATION IN THE LIMIT

We next present our results on robust identification in the limit. Our results demonstrate a trichotomy on the tolerance of error. In Section 4.1, we prove that for DFAs, robust identification can be done in the constant error-rate regime. In Section 4.2, we show that robust identification is impossible for DPDAs in the constant error rate regime, but plausible in the diminishing error regime. Finally, for the set of TMs, robust identification is impossible even with diminishing error, but for a finite number of errors, there is a robust identification algorithm; see Section 4.3.

**Overview of technique** At a high level, the algorithms in this Section are very different from the algorithm in section 3. Instead of reconstructing the state transition from the (corrupted) traces, we use them as evidence on the target function class. In particular, we maintain the set of states that are observed in the computational trace and we prove that either (1) the number of states keeps growing, or (2) the target function class contains not so many states (compared to the observed states). The first case can only occur for a finite number of times so eventually we can reduce the task to finite class language identification. All algorithms in this section follow the same template, and the main technical step is to establish the size bound of (2). While this is not so hard for DFAs and TMs, the size bound for DPDAs is quite tricky and we need to analyze DPDAs through the Chomsky Normal Form (CNF) decomposition and bound the size of the corresponding CNF tree. Due to space constraints, we only sketch the main idea of the proofs and the formal proofs can be found at Appendix B.

### 4.1 DETERMINISTIC FINITE AUTOMATA

**Theorem 4.1.** *For the task of robust identification in the limit with the set of machines as DFAs, there is a robust identification algorithm that guarantees to succeed in the constant error rate regime.*

**Notation** We use $M^*$ to denote the target DFA and $Q^*$ the set of states in $M^*$. Let $Q^*_{\text{accept}} \subseteq Q^*$ be the set of states that have ever appeared in the computation trace of an accepted string $x \in L(M^*)$. Without loss of generality, we assume the alphabet size is 2 ($|\Sigma| = 2$).

**Algorithm.** At time $t$, let $Q_t$ be the set states that appear in the noisy traces $\widetilde{c}_{M^*}(x_1), \ldots, \widetilde{c}_{M^*}(x_t)$. Let

$$B_t := |Q_t| \cdot 2^{2|Q_t|/(1-\alpha)+\ell_\alpha} + 1.$$

At step $t$ the algorithm outputs a minimal DFA $M_t$ that is (i) consistent with the history $x_1, \ldots, x_t$ (i.e., $x_i \in L(M_t)$ for all $i \leq t$) and (ii) has at most $B_t$ states.

We state a few simple facts. First, the set of $Q_t$ must stabilize after some finite time step.

**Fact 4.1.** *There exists $t'$ such that $Q_t = Q_{t'}$ for all $t \geq t'$.*

Fix $t'$ from Fact 4.1 and define $U := Q_{t'}$. For any accepted string $x \in L(M^*)$ with trace $c_{M^*}(x)$, write

$$m(x) := \big|\{j : c_{M^*}(x)_j \notin U\}\big|.$$

That is, $m(x)$ is the number of states that are not in $U$.

**Fact 4.2.** *For every accepted $x \in L(M^*)$ with length $|x| \geq \ell_\alpha$ we have $m(x) \leq \alpha|x|$.*

For a state $q \in Q^*$, define its graph distance to $U$ (along DFA transitions) as

$$\mathrm{dist}_U(q) := \min\{m \geq 0 : \exists \, q' \in U \text{ and a path of length } m \text{ from } q' \text{ to } q\},$$

The key step is to prove that every accepting state is not far from the set $U = Q_{t'}$.

**Lemma 4.2.** *For any state $q \in Q^*_{\mathsf{accept}}$, we have*

$$\mathrm{dist}_U(q) \leq \max\left\{\frac{2}{(1-\alpha)} \cdot |U|, \ell_\alpha\right\}.$$

*Proof of Theorem 4.1.* By Lemma 4.2, every $q \in Q^*_{\mathsf{accept}}$ lies within distance at most $\frac{2}{(1-\alpha)}|U| + \ell_\alpha$ from $U = Q_{t'}$. We have assumed the alphabet size $|\Sigma| = 2$, so the size of $Q^*_{\mathsf{accept}}$ is at most

$$|Q^*_{\mathsf{accept}}| \leq |Q_{t'}| \cdot 2^{\frac{2}{(1-\alpha)}|Q_{t'}|+\ell_\alpha} = |B_{t'}| - 1.$$

Therefore, from time $t'$ onward, the hypothesis class considered by the algorithm is a fixed finite family that contains $L(M^*)$. Applying the finite-class identification argument (Fact 2.1), the algorithm will identify the target language in the limit. This completes the proof. $\qquad\square$

## 4.2 DETERMINISTIC PUSHDOWN AUTOMATA

**Theorem 4.3.** *Consider the task of robust language identification in the limit with the set machines as DPDAs, then*

- *There is an algorithm that guarantees to succeed in the diminishing error rate regime;*

- *Robust language identification is impossible in the constant error rate regime, for any constant $\alpha > 0$.*

**Notation** We use $M^*$ to denote the target DPDA and $Q^*$ the set of states in $M^*$. Let $Q^*_{\mathsf{accept}} \subseteq Q^*$ be the set of states that have ever appeared in the computational trace of an accepted string $x \in L(M^*)$. We assume the alphabet for input string and stack string are binary. For each operation of DPDA, it reads one input symbol, then pushes or pops or pushes-then-pops one symbol on top of the stack. A string is accepted if and only if the stack is empty at the end – for this to be true, we allow the DPDA to pop stack symbols when it reaches to the end of the string.

**Remark 1** (DPDA normal form). *Throughout Section 4.2, we assume DPDAs are in a normal form where each transition (1) reads either one input symbol or $\epsilon$, and (2) modifies the stack by at most one symbol (push or pop one symbol). We also assume acceptance by empty stack. These assumptions ensure that for fixed alphabets and a state bound, the induced hypothesis class is finite, and they can be enforced by a standard conversion at the cost of a polynomial blow-up in the number of states.*

**Algorithm** Let $Q_t$ be the set of states observed in the traces $\widetilde{c}_{M^*}(x_1), \ldots, \widetilde{c}_{M^*}(x_t)$ up to time $t$. Let $\alpha_t = 2^{-2|Q_t|^6}$. Let $\mathcal{M}_t$ be the set of DPDAs with number of states at most

$$B_t = |Q_t| \cdot \left( 32^{2^{|Q_t|^7 + \ell_{\alpha_t}}} \right) + 1$$

The algorithm outputs the minimal DPDA $M_t \in \mathcal{M}_t$ that accepts $x_1, \ldots, x_t$. As before, we show:

**Fact 4.3.** *The set $Q_t$ is monotone non-decreasing and $Q_t \subseteq Q^*$. Therefore, there exists time $t_0$, such that $Q_t = Q_{t_0}$ for any $t \geq t_0$.*

We write $S = Q_{t_0}$, $s = |Q_{t_0}|$ and $\alpha = \alpha_{t_0} = 2^{-2s^6}$ from now on.

**Definition 6** (Transition graph). *Define the transition graph with the node set being the state set $Q^*$. There is a directed edge from a state $q \in Q^*$ to another state $q' \in Q^*$, if and only if there is a one step transition from $q$ to $q'$, i.e., there exists input symbol $i_1 \in [2]$, top stack symbol $i_2 \in [2]$, operation $j \in [3]$, write symbol $u \in [2]$, such that $\delta(q, i_1, i_2) = (q', j, u)$. For any state $q \in Q^*_{\text{accept}}$, define its distance $\mathrm{dist}_S(q)$ to $S$ as the length of the minimum path that starts from a state in $S$ and ends at $q$.*

Our main result here is as follows.

**Lemma 4.4.** *For any state $q \in Q^*_{\text{accept}}$, one has*

$$\mathrm{dist}_S(q) \leq \max \left\{ \ell_\alpha, 2^{s^7} \right\}.$$

The proof of Lemma 4.4 can be found at Appendix B.2. The key idea is to convert DPDA computation trace to its corresponding Chomsky Normal Form (CNF), and then prove that the CNF tree can not be too large, otherwise there are too many unobserved states in the CNF tree.

*Proof Sketch of Theorem 4.3.* We first prove the correctness of our algorithm in the diminishing error regime. By Lemma 4.4, all acceptable states are at most $\max \left\{ \ell_\alpha, 2^{s^7} \right\}$ far from the set $S$. By the definition of the transition graph (Definition 6), each state has constant out degree (actually at most 32), so the total number of acceptable states is bounded by $B_t - 1$. The correctness of our algorithm then follows directly from Fact 2.1 for finite class language identification.

In the constant error rate regime, fix any constant $\alpha \in (0, 1)$, and consider the following set of languages $\mathcal{L} = \cup_{i \in \mathbb{N}} L_i \cup L_\infty$, where for any $i \in \mathbb{N}$, $L_i := \{a^n b^{2n/\alpha}, n \leq i\}$ and $L_\infty := \{a^n b^{2n/\alpha}, n \in \mathbb{N}\}$. It is possible to construct DPDAs $\{M_i\}_{i \in \mathbb{N}} \cup M_\infty$ such that $M_i$ recognizes $L_i$ ($i \in \mathbb{N} \cup \{\infty\}$). For any string $x = a^n b^{2n/\alpha}$, the adversary would remove the first $n$ steps such that the computational trace reveals zero information regarding the core mechanism of $\{M_i\}_{i \in \mathbb{N}} \cup M_\infty$ except the membership of $x$. Then, from the learner's perspective, the problem essentially reduces to the hard instance of Fact 2.2. $\square$

### 4.3 TURING MACHINES

**Theorem 4.5.** *Consider the task of robust language identification in the limit with the set machines as Turing machines, then*

- *There is an algorithm that guarantees to succeed when the number of errors is finite;*

- *Robust language identification is impossible even in the diminishing error regime.*

The algorithm, analysis and the hard instance for Theorem 4.5 are quite similar to Theorem 4.3; due to space constraints we defer the full proof to Appendix B.3.

## 5 CONCLUSION AND FUTURE WORK

In this work we have studied the problem of identification in the limit augmented with a notion of chain of thought. Our results show a strong theoretical benefit of utilizing CoT: while in Gold's

original setting almost every interesting collection of languages is *not* identifiable in the limit, in our setting *all* collections of languages that are recognizable by Turing machines are identifiable in the limit with CoT. Moreover, our results show that, depending on the complexity of the machines that underlie the collection of languages, there are identification algorithms that can handle varying amounts of noise in the CoT. There are several interesting follow-up questions related to our work. Firstly, our results are asymptotic in nature. It is, thus, natural to try to find conditions under which one can prove quantitative bounds on the amount of time that is needed in order to identify the target language with CoT information. A similar question can be asked in the setting of *generation* in the limit. While Kleinberg & Mullainathan (2024) showed that, asymptotically, every collection is generatable, it is known that achieving correct generation might take arbitrarily long for several collections (Li et al., 2024). Hence, it would be interesting to see how CoT information can speed up the generation process. Relatedly, several works have studied trade-offs between generation with breadth and hallucinations Kleinberg & Mullainathan (2024); Kalavasis et al. (2024b;a); Charikar & Pabbaraju (2024); Peale et al. (2025); Kleinberg & Wei (2025a). It is intriguing to understand how the landscape of these trade-offs changes in the presence of CoT information.

## ACKNOWLEDGEMENTS

This work was supported in part by AFSOR grant FA9550-23-1-0251 and ONR grant N000142212771.

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

## A  OMITTED PROOF FROM SECTION 3

Below, we give the proof of Theorem 3.1.

*Proof of Theorem 3.1.* We now show that access to perfect traces suffices to identify in the limit any language recognized by a *deterministic* Turing machine. We maintain a working TM $M_t = (Q_t, \Sigma, \Gamma_t, \delta_t, q_0, B, F_t)$ together with a designated non-accepting *reject sink* state $r$. Intuitively, any transition entry not yet witnessed on an accepting trace is pessimistically set to jump to $r$; witnessed entries are overwritten to match $\delta^*$.

**Initialization.** Set $Q_0 = \{q_0, r\}$, $F_0 = \emptyset$, and $\Gamma_0 = \{B\} \cup \Sigma$. Define $\delta_0(q, \gamma) = (r, \gamma, R)$ for all $q \in Q_0$ and $\gamma \in \Gamma_0$ (and, say, $\delta_0(r, \gamma) = (r, \gamma, R)$ to keep $\delta_0$ total).

**Update on a positive example** $(x_t, c(x_t))$**.** Parse the trace and:

1. Add to $Q_t$ every state that appears in $c(x_t)$; add to $\Gamma_t$ every tape symbol that appears in $c(x_t)$.

2. If the last configuration is in some accepting state $q^{\text{acc}}$, set $F_t \leftarrow F_{t-1} \cup \{q^{\text{acc}}\}$.

3. For each revealed step $(q, \gamma) \mapsto (q', \gamma', D)$ of the trace, *overwrite* $\delta_t(q, \gamma) \leftarrow (q', \gamma', D)$. For any newly added pair $(q, \gamma)$ still undefined after this pass, set $\delta_t(q, \gamma) = (r, \gamma, R)$ to keep $\delta_t$ total.

Determinism of $M^*$ implies no conflicts can arise when overwriting previously set entries.

**Subset invariant.** For all $t$, $L(M_t) \subseteq K$. Indeed, any accepting run of $M_t$ uses only (i) transitions copied verbatim from $\delta^*$ because they appeared on some accepting trace, or (ii) default entries to $r$, which cannot lie on an accepting computation. Hence $M_t$ never accepts a string outside $K$.

**Eventual completeness on the accepting support.** Let

$$S^+ = \left\{ (q, \gamma) \in Q^* \times \Gamma^* : \delta^*(q, \gamma) \text{ is used on at least one accepting computation of } M^* \right\}.$$

Since $Q^*$ and $\Gamma^*$ are finite, $S^+$ is finite. For every $(q, \gamma) \in S^+$ fix a witness string $w_{(q,\gamma)} \in K$ whose accepting trace uses $(q, \gamma)$. Each $w_{(q,\gamma)}$ appears at some finite time, after which the update step overwrites $\delta_t(q, \gamma)$ to $\delta^*(q, \gamma)$. Let $t^\star$ be the latest such time over all $(q, \gamma) \in S^+$. Then for all $t \geq t^\star$:

$$\delta_t(q, \gamma) = \delta^*(q, \gamma) \quad \text{for every } (q, \gamma) \in S^+,$$

and all states/symbols/accepting states that ever occur on accepting runs have been added to $Q_t, \Gamma_t, F_t$.

**Convergence.** Fix any $x \in K$ and consider its accepting computation under $M^*$. Every step of that computation uses an entry in $S^+$. Therefore, for $t \geq t^\star$, the same sequence of steps is a valid accepting computation under $M_t$, yielding $K \subseteq L(M_t)$. Combined with the subset invariant, we obtain $L(M_t) = K$ for all $t \geq t^\star$. No further changes occur after $t^\star$ (determinism prevents conflicting overwrites), so the hypothesis stabilizes, as required by Definitions 1 and 3. $\qquad\square$

## B  OMITTED PROOF FROM SECTION 4

### B.1  OMITTED PROOF FROM SECTION 4.1

*Proof of Fact 4.1.* The sequence $Q_1 \subseteq Q_2 \subseteq \cdots$ is monotone increasing and bounded above by the finite set $Q^*$; hence it stabilizes. $\qquad\square$

*Proof of Fact 4.2.* The edit distance between $\widetilde{c}_{M^*}(x)$ and the actual trace $c_{M^*}(x)$ is at most $\alpha|x|$. By the definition of $t'$, $\widetilde{c}_{M^*}(x)$ only contains states in $U$, while there are $m(x)$ states of $c_{M^*}(x)$ that are not in $U$. By the finite error rate assumption, we must have $m(x) \leq \alpha|x|$. $\qquad\square$

*Proof of Lemma 4.2.* Consider any state $q \in Q^*_{\text{accept}}$ and take the shortest accepted string $x$. It suffices to consider the case that $|x| \geq |\ell_\alpha|$ (otherwise $\text{dist}_U(q) \leq \ell_\alpha$). Consider its computation trace $c_{M^*}(x) := c_{\text{pre}} \circ q \circ c_{\text{suf}}$, both the prefix $c_{\text{pre}}$ and the suffix $c_{\text{suf}}$ of the position $q$ is a simple path (otherwise the string can be shortened), therefore, one has

$$|\{j : c_{M^*}(x)_j \in U\}| \leq 2|U|.$$

By Fact 4.2, we have

$$\alpha \geq \frac{m(x)}{|x|} \geq \frac{|x| - 2|U|}{|x|} \quad \Rightarrow \quad |x| \leq \frac{2}{1 - \alpha} \cdot |U|$$

Hence, we have $\text{dist}_U(q) \leq |x| \leq \frac{2}{(1-\alpha)} \cdot |U|$, this completes the proof. $\square$

## B.2 OMITTED PROOF FROM SECTION 4.2

**Definition 7** (CNF tree). *For any two states $p, q \in Q^*$, we introduce a non-terminal $A_{p,q}$ intended to generate precisely the strings that the DPDA reads, starts with state $p$, ends with state $q$, and leaving the original stack untouched. This is similar to the standard conversion to Chomsky normal form and generates grammar rules of*

- $A_{p,q} \rightarrow A_{p,r} A_{r,q}$, $r \in Q^*$

- $A_{p,q} \rightarrow a A_{r,s} b$, $a \in [2], b \in [2] \cup \{\epsilon\}$

*Given an accepted string $x$, we can generate a unique trinary tree according to the the grammar, and we denote this tree as $\Gamma(x)$.*

*Proof of Lemma 4.4.* Let $v \in Q_{\text{accept}}$ be any acceptable state. Among all accepted strings whose computation visits the state $v$, pick one of minimum length and denote the string by $x$. If $|x| \leq \ell_\alpha$, then it holds trivially that $\text{dist}_S(v) \leq \ell_\alpha$. Hence, it suffices to consider the case that $|x| \geq \ell_\alpha$. By our assumption the edit distance between $c_{M^*}(x)$ and $\widetilde{c}_{M^*}(x)$ is at most $\alpha|c_{M^*}(x)|$. Let $\Gamma = \Gamma(x)$ be the CNF derivation tree of string $x$ under Definition 7. Define the *principal path* as any root-to-internal-node path ending at the node $N_v$ labeled by some $A_{p,q}$ with $v \in \{p, q\}$. Let $H$ be the number of internal nodes on this path and $|\Gamma|$ be the size of the tree.

**Lemma B.1** (Non repetition). *Consider any path $\mathcal{P}$ in the tree (not necessary from leaf to the root), then there is no repetition for $\mathcal{P}$, if*

- *the path $\mathcal{P}$ has no intersection with the principal path*

- *$\mathcal{P}$ is the principal path.*

*Here no repetition means for any $A_{p,q}, A_{p',q'} \in P$, $A_{p,q} \neq A_{p',q'}$.*

*Proof.* We prove by contradiction. Suppose there are two states $A_{p,q} = A_{p',q'}$, $A_{p,q}, A_{p',q'} \in \mathcal{P}$, then consider the string $x'$ that removes the $A_{p,q} \setminus A_{p',q'}$ (suppose $A_{p,q}$ is closer to the root), $x'$ is an accept string shorter than $x$, moreover, $A_{p,q} \setminus A_{p',q'}$ does not contain $N_v$. This is because, (1) when $\mathcal{P}$ is the principal path, $N_v \in A_{p',q'}$, (2) when $\mathcal{P}$ has no intersection with the principal path, $N_v \notin A_{p,q}$. Therefore, the string $x'$ is shorter than $x$ and also contains the state $v$. This is a contradiction. $\square$

We now treat two cases based on the size of $\Gamma$.

**Case 1. The tree is relatively small comparing to the principal path:** $|\Gamma| \le 2^{s^6} H$**.** By Lemma B.1, along the principal path, at most $s^2$ nodes can be labeled by variables $A_{p,q}$ whose *both* endpoints $p, q$ lie in $S$; therefore at least $H - s^2$ nodes involve a new state in their endpoints. Here we say a state $q$ is a new state if $q \in Q^*_{\text{accept}} \setminus S$. Each such node forces a visit to a new state in the accepting computation.

If $H > s^3$ then,

$$\frac{\#\text{ new-state visits}}{|c_{M^*}(x)|} \ge \frac{H - s^2}{2|\Gamma|} \ge \frac{H/2}{2 \cdot 2^{s^6} H} > 2^{-2s^6} = \alpha,$$

which contradicts the fact that $\widetilde{c}_{M^*}(x)$ has at most $\alpha|c_{M^*}(x)|$ edit distance error, as it is impossible to remove all these new states.

Therefore, in case 1, we must have $H \le s^3$, and thus

$$\text{dist}_S(v) \le |x| \le |\Gamma| \le 2^{s^6} H \le 2^{s^6} s^3 \le 2^{s^7}$$

**Case 2. The tree is relatively large comparing to the principal path:** $|\Gamma| > 2^{s^6} H$. We prove this can not happen. Call a leaf *off-path* if its distance to the principal path is at least $d = s^5$. In a trinary tree, the number of leaves within distance $d$ of a path of length $H$ is at most $3^d H$, hence the number of off-path leaves is at least

$$\frac{1}{2}|\Gamma| - 3^d H = \frac{1}{2}|\Gamma| - 3^{s^5} H > \frac{1}{2}|\Gamma| - 3^{s^5} \cdot 2^{-s^6}|\Gamma| \ge \frac{1}{4}|\Gamma|. \tag{1}$$

For each off-path leaf $u$, let $\pi(u)$ be the truncated chain of the last $d$ edges on the leaf-to-root path. By definition $\pi(u)$ is disjoint from the principal path. By Lemma B.1, there is no repetition on any $\pi(u)$. Along $\pi(u)$, at most $s^2$ nodes have both endpoints in $S$, hence at least $d - s^2 \ge d/2 = s^5/2$ nodes involve a new state.

By Eq. equation 1, there are at least $\frac{1}{4}|\Gamma|$ such length $d$ path, each path contains $s^5/2$ new states, and each states can be counted for at most $3^d = 3^{s^5}$ times, so there are at least

$$\frac{1}{4}|\Gamma| \cdot s^5/2 \cdot 3^{-s^5} \ge 4^{-s^5}|\Gamma| \ge \alpha|x|$$

new states appear in the computation trace, this violates the edit distance error assumption. Hence this case can not happen.

Combining the above two cases, we conclude that $\text{dist}_S(q) \le \max\{\ell_\alpha, 2^{s^7}\}$ and complete the proof. $\square$

*Proof of Theorem 4.3, Part 1.* By Lemma 4.4, every state $q \in Q^*_{\text{accept}}$ lies within distance at most $\ell_\alpha + 2^{s^7}$ from the set $Q_{t_0}$. By the definition of the transition graph (see Definition 6), the out-degree for each state is at most 32, hence, we have

$$|Q^*_{\text{accept}}| \le |S| \cdot 32^{\ell_\alpha + 2^{s^7}} = |Q_{t_0}| \cdot 32^{\ell_\alpha + 2^{|Q_{t_0}|^7}} = B_{t_0} - 1$$

where the second step follows from the definition that $S = Q_{t_0}$ and $s = |S|$. Therefore, from time $t_0$ onward, the hypothesis class considered by the algorithm is a fixed finite family that contains $L(M^*)$. Applying the finite-class identification argument (Fact 2.1), the algorithm would identify the target set of languages in the limit. This completes the proof. $\square$

We next prove that robust language identification is impossible for DPDAs in the constant error regime.

*Proof of Theorem 4.3, Part 2.* Fix any constant $\alpha \in (0, 1)$, consider the following set of language $\mathcal{L} = \cup_{i \in \mathbb{N}} K_i \cup K_\infty$, where for any $i \in \mathbb{N}$,

$$K_i := \{a^n b^{2n/\alpha}, n \le i\}$$

and

$$K_\infty := \{a^n b^{2n/\alpha}, n \in \mathbb{N}\}$$

Consider the set of DPDAs $\mathcal{M} = \{M_i\}_{i \in \mathbb{N}} \cup M_\infty$ and we would construct them such that $M_i$ recognize exactly the set of language $K_i$ ($i \in \mathbb{N} \cup \{\infty\}$).

For $i \in \mathbb{N}$, the DPDA $M_i$ has $i + 2/\alpha + O(1)$ states. Given an input string $x$, it first reads all the $a$ symbol in the beginning. It pushes all $a$ symbol to the stack, and at the same time, it uses the first $i$ states to perform counting. If the number of $a$ is greater than $i$, then it goes to a reject state after reading the $(i + 1)$-th 1. Upon reading the first 0 symbol, the DPDA moves to a temporary accept state if the number of $a$ it has read is at most $i$. After this, the DPDA starts to pop out the $a$ symbol in the stack. In particular, it pops out one $a$ symbol after reading $2/\alpha$ symbol $b$ (counting can be done using $2/\alpha$ states). The DPDA $M_i$ would accept the string if it is in the temporary accept state and the stack is empty at the end.

The DPDA $M_\infty$ has $2/\alpha + O(1)$ states. It has the same transition function except it is always in the temporary accept state when reading the $a$ symbol at the beginning.

It is easy to prove that $M_i$ recognize exactly the set $K_i$, i.e., $K_i = \mathcal{L}(M_i)$. For the corruption, given a string $x = a^n b^{2n/\alpha}$, the adversary would remove the first $n$ state transitions, so the algorithm sees only the popping operations starting from the temporary accept state or the reject state. The number of deletion is $n$, so the edit distance error rate between the corrupt computation trace and the actual trace is $n/(n + 2n/\alpha) < \alpha$. The corrupted string reveals zero information regarding the DPDA, except the membership of the string. Hence, by a simple reduction from the hard instance of Fact 2.2, it is impossible for language identification in the constant error rate regime. $\qquad\square$

### B.3 Omitted proof from Section 4.3

**Notation** We use $M^*$ to denote the target Turing machine, $Q^*$ be the set of state in $M^*$. Let $Q^*_{\text{accept}} \subseteq Q^*$ be the set of states that have ever appeared in the computation trace of an accepted string $x \in L(M^*)$. We assume the alphabet for both input string and the writing tape are binary. We assume Turing machine has an input tape (read-only) and one write-tape, our results extend to multiple write-tapes easily. The transition function $\delta$ takes an input state $q$, tape value $r, w \in [2]$, and outputs the transition state $q'$, head movement $m_1, m_2 \in \{-1, 0, 1\}$ and write value $w' \in [2]$, a.k.a. $\delta : Q \times [2] \times [2] \to Q \times [3] \times [3] \times [2]$.

**Algorithm** Recall in the finite-error regime, there exist a pair of constants $C \in \mathbb{N}$ and $\ell_C \in \mathbb{N}$, such that for any example $(x, \widetilde{c}_{M^*})$ with input length at least $|x| \geq \ell_C$, the edit distance between the actual trace $c_{M^*}(x)$ and the corrupted trace $\widetilde{c}_{M^*}(x)$ is at most $C$.

For any time step $t$, let $Q_t$ be the set of states that appears in the noisy traces $\widetilde{c}_{M^*}(x_1), \ldots, \widetilde{c}_{M^*}(x_t)$. Let $\mathcal{M}_t$ be the set of Turing machines that have at most

$$B_t = |Q_t| \cdot 72^{C+1+\ell_C} + 1$$

states . The algorithm at step $t$ selects the minimal Turing machine in $\mathcal{M}_t$ that accepts the strings $x_1, \ldots, x_t$.

Similar as before, we have

**Fact B.1.** *The set $Q_t$ is monotone non-decreasing and $Q_t \subseteq Q^*$. Therefore, there exists time $t_0$, such that $Q_t = Q_{t_0}$ for any $t \geq t_0$.*

From now on, we write $S = Q_{t_0}$, $s = |Q_{t_0}|$. Next, we define the state transition graph over states $Q^*$. For any state $q, q' \in Q^*$, there is a directed edge from $q$ to $q'$ if and only if there is a one step transition from $q$ to $q'$. More formally, there is an edge from $q$ to $q'$ if there exists $r, w \in [2]$, $m_1, m_2 \in [3]$, $w' \in [2]$, such that $\delta(q, r, w) = (q', m_1, m_2, w')$. For any state $q \in Q^*_{\text{accept}}$, define its distance to $S$ as the length of the minimum path that starts from a state in $S$ and ends in $q$, i.e.,

$$\text{dist}_S(q) := \min\{m \geq 0 : \exists\, q_0 \to q_1 \to \cdots \to q_m = q,\ q_0 \in U,\ q_1, \ldots, q_m \notin U\},$$

We have

**Lemma B.2.** *For any $q \in Q^*_{\text{accept}}$, we have*

$$\text{dist}_S(q) \leq \max\{\ell_C, C\}.$$

*Proof.* Consider an arbitrary accept string $x$ whose computation path visits the state $q$. If the computation path $c_{M^*}(x) \leq \ell_C$, then we already have $\text{dist}_S(q) \leq \ell_C$. On the other hand, if $c_{M^*}(x) \geq \ell_C$, we have the edit distance between $\widetilde{c}_{M^*}(x)$ and $c_{M^*}(x)$ is at most $C$. We note that all states that appear in $\widetilde{c}_{M^*}(x)$ must be in $S$, hence there are at most $C$ states in $c_{M^*}(x)$ that are not in $S$ (due to the bound on edit distance). This implies that $\text{dist}_S(q) \leq C$ and completes the proof. $\qquad \square$

Now we can wrap up the first part of Theorem 4.5.

*Proof of Theorem 4.5, Part 1.* By Lemma B.2, every state $q \in Q^*_{\text{accept}}$ lies within distance at most $\ell_C + C$ from the set $Q_{t_0}$. By the definition of the transition graph, the out-degree for each state is at most 72, hence, we have

$$|Q^*_{\text{accept}}| \leq |S| \cdot 72^{\ell_C + C + 1} = |Q_{t_0}| \cdot 72^{\ell_C + C + 1} = B_{t_0} - 1$$

Therefore, from time $t_0$ onward, the hypothesis class considered by the algorithm is a fixed finite family that contains $L(M^*)$. Applying the finite-class identification argument (Fact 2.1), the algorithm would identify the target set of language in the limit. This completes the proof. $\qquad \square$

Next, we prove that robust language identification for TM is impossible in the diminishing error regime

*Proof of Theorem 4.5, Part 2.* We construct a hard family of target languages and Turing machines. For $i \in \mathbb{N}$, define $K_i := \{a, aa, \ldots, a^i\}$ and $K_\infty := a^*$. Consider the following set of Turing machines $\{M_i\}_{i \in \mathbb{N} \cup \{\infty\}}$. We would use a padding approach. For any $i \in \mathbb{N}$, the Turing machine $M_i$ has $i + O(1)$ states. It first counts the number of $a$ in the input string and compares with $i$, if the number of $a$ is greater than $i$, then it enters a reject state, otherwise, it enters the accept states. This takes at most $|x|$ steps in total. After this, $M_i$ would cycle in this state for $|x|^2$ steps. The definition of $M_\infty$ is similar, except it skips the first stage, it always enter the accept state (as long as all symbol in the input are $a$).

It is easy to see that $K_i$ is exactly the set of string accepted by $M_i$, i.e., $K_i = L(M_i)$ ($i \in \mathbb{N} \cup \{\infty\}$). Consider the following adversary, for each accepted string, the corrupted trace $\widetilde{c}_{M^*}(x)$ would delete all state transition in the first stage and contain only $|x|^2$ cycle steps. We note the corrupt ratio is $1/|x|$ the error rate is diminishing. On the other hand, the corrupted trace $\widetilde{c}_{M^*}(x)$ reveals zero information about the state transitions of the TM, except the membership of $x$. Hence, by a simple reduction from the hard instance of Fact 2.2, it is impossible for robust language identification in the diminishing error regime. $\qquad \square$

