# OpenReview forum: "Language Identification in the Limit with Computational Trace"
_ICLR.cc/2026/Conference — ICLR 2026 Poster_

### Official Review · Reviewer_XJpB · 2025-10-21

**Soundness:** 3
**Presentation:** 3
**Contribution:** 3
**Rating:** 6
**Confidence:** 3

**Summary:**

The paper extends Gold’s seminal formal language learning framework from learning only based on positive examples to learning based on positive examples and computational traces associated with them. This is relevant to modern language models as chain-of-thought traces have been theoretically linked to such execution traces, making the setting useful to study learnability of algorithms.
The authors find that, in stark contrast to Gold’s result, traces make large classes of languages, including all recursively-enumerable languages, learnable. The authors then study learnability under corrupted traces, and find that it makes the learning problem markedly harder; while regular languages remain learnable under a constant fraction of errors, context-free languages and Turing machines require much stricter restrictions on the corruption.

**Strengths:**

- I believe the paper studies a very interesting and useful problem and puts a new spin on an old/classic setting
	- In particular, it provides another possible contributing factor behind how CoT helps improve models
- The exposition and motivation are clear; it’s easy to discern what the paper’s contributions are, andthe  methodology was used/developed
	- For example, the proofs are first well-described on simpler models (finite-state automata)
	- The related work section is thorough and useful

**Weaknesses:**

- Although this is not a major drawback, I feel like the connection to generation in the limit, which first appears in the Introduction, is not really justified; I’m not sure how learning with traces is any more connected to generation in the limit than the original learning in the limit setting
- Very minor, and I don’t think this undermines the theoretical contributions: It is slightly unclear how the results translate into practice; maybe at least describing how this could be used or tested in practice could be useful

**Questions:**

- As you mention, the results are asymptotic in nature, which is okay. I was just wondering if you have any ideas for next steps, i.e., how one would proceed/extend the results to some complexity bounds? I imagine the methodology would have to be quite different.
- Can you elaborate on the connection to generation in the limit? It seems like, since there are fewer impossibility results there, traces would not be as useful?

---

> ### Author Response · Authors · 2025-11-21
>
> We thank the reviewer for their time and constructive feedback. We are pleased to see that they found the problem interesting and our exposition clear. We address their specific comments and questions below.
>
> >Although this is not a major drawback, I feel like the connection to generation in the limit, which first appears in the Introduction, is not really justified; I’m not sure how learning with traces is any more connected to generation in the limit than the original learning in the limit setting.
>
> We agree with the reviewer that our setting, as treated in this paper, is much closer to identification in the limit than generation in the limit. We cited the latter primarily to contextualize our motivation: that line of work served as a catalyst for revitalizing interest in asymptotic, limit-style analysis as a rigorous lens for understanding LLMs. That said, we believe traces are relevant to the generation setting as well. While Kleinberg and Mullainathan (2024) showed that generation in the limit is always possible, they did not quantify the sample complexity. Subsequent work by Li, Raman, and Tewari (2024) introduced notions of uniform and non-uniform generation to capture the speed of convergence. Their results show that not all countable collections are uniformly generatable—meaning an adversary can delay the generator's success arbitrarily. An exciting direction for future work is to determine if access to computational traces can bridge this gap, converting non-uniformly generatable classes into uniformly generatable ones, or significantly improving the computational efficiency of the generating algorithms.
>
> >Very minor, and I don’t think this undermines the theoretical contributions: It is slightly unclear how the results translate into practice; maybe at least describing how this could be used or tested in practice could be useful.
>
> Thank you for this suggestion. We see two primary ways to translate these results into practice:
> 1. Our theory suggests that access to intermediate steps (CoT type of annotation) is crucial for language identification and learning. Since language identification can be thought of as building some type of a world model, this suggests that LLMs could benefit from CoT type of information in order to build such world models.
> 2. Our theory suggests that for complex and harder tasks higher quality of trace/CoT might be more critical as our hierarchical separations for the noisy setting show learning automata is quite robust to noisy CoT, whereas learning TMs can tolerate much less noise. Of course, these claims require extensive and rigorous empirical justification and we believe these are exciting future directions that our work can inspire.
>
> >As you mention, the results are asymptotic in nature, which is okay. I was just wondering if you have any ideas for next steps, i.e., how one would proceed/extend the results to some complexity bounds? I imagine the methodology would have to be quite different.
>
> This is also a very good point. We view our asymptotic results as the first step towards studying this setting, and we believe an interesting future direction is to derive non-asymptotic bounds. A starting point would be to use a notion of “uniform identification” (inspired by the work of Li, Raman, and Tewari on uniform generation), which asks for algorithms that identify the target after seeing at most d examples (with their traces), where d doesn’t depend on the target or the enumeration. We suspect that not all languages would satisfy this notion of learnability, but it would be interesting to see what properties of the collection are necessary to enable that.
>
> >Can you elaborate on the connection to generation in the limit? It seems like, since there are fewer impossibility results there, traces would not be as useful?
>
> Please see our comment above.

---

> > ### Comment · Reviewer_XJpB · 2025-11-24
> >
> > Thank you for the clarifications!
> > The explanation of the connection to generation in the limit makes more sense now, yes. I believe including a sentence or two about this in the paper would be beneficial.
> > I maintain my positive evaluation of the paper and hope it is accepted.

---

> > > ### Author Response · Authors · 2025-11-25
> > >
> > > Thank you for taking the time to read our rebuttal and for supporting our paper. We will make sure to include this discussion to the next version of our work.

---

### Official Review · Reviewer_WbPo · 2025-10-27

**Soundness:** 3
**Presentation:** 3
**Contribution:** 3
**Rating:** 8
**Confidence:** 3

**Summary:**

This paper proves theoretical results concerning language identification in the limit in the case when the learner is given an enumeration of positive strings plus the computational traces that demonstrate that an automaton for the language to be identified accepts each string (e.g., a sequence of states in a finite automaton). The main result of the paper is this: unlike the classical case consisting only of positive strings  without computational traces, in which case most interesting language classes are not identifiable, the class of computable languages is identifiable in the limit when computational traces are provided. The authors then prove results for cases when the computational traces contain a certain number of errors for the classes of regular languages, deterministic context-free languages, and computable languages.

**Strengths:**

Overall, this is an interesting paper with significant, original theoretical results. As the authors point out, these results are relevant to the use of CoT to train LLMs. The paper is written clearly and does a good job of contextualizing itself amid prior work. The robustness results are quite interesting.

**Weaknesses:**

The paper would benefit from some clarifications; see my comments in the Questions section.

1. A minor point, but I would point out in the abstract that you are assuming that the learner only has access to positive examples, not negative examples.
1. 072: DPDAs correspond to DCFLs, not CFLs.
1. 183: The definition of DPDA is missing constraints on $\delta$ that make it deterministic. I think you need to allow $\varepsilon$ as the popped symbol too.

**Questions:**

1. 088: Do you mean that the number of errors per trace is $O(1)$ with respect to length, not finite?
1. Fact 2.2: In this example, what is the alphabet $\Sigma$?
1. Theorem 3.1: Do we assume all the TMs are deciders?
1. 310: Does this work if the alphabet is not fixed to {0, 1} ahead of time?
1. 380: How is it possible for $m(x)$ not to be 0? $U$ is already the set of all states that occur in all traces. Can different instances of the traces for the same $x$ be edited in different ways, or are they always consistent?
1. 388: This statement doesn't make sense to me. Do you mean "not every accepting state is from the set $U$"?
1. Def 6: What about non-scanning transitions?

Typos:
1. 267: the the
1. 371: set states -> set of states

---

> ### Author Response · Authors · 2025-11-21
>
> We thank the reviewer for their time and for offering very useful feedback and comments. We are happy to see that the reviewer found our theoretical results significant and relevant to practical applications. We address the specific comments below and will revise our manuscript to incorporate these discussions. We will also fix the typos you pointed out; thanks for catching them.
>
> >A minor point, but I would point out in the abstract that you are assuming that the learner only has access to positive examples, not negative examples.
>
> Thanks for the suggestion, we will do that. Indeed, having access to only positive examples is what drives Gold’s result.
>
> >072: DPDAs correspond to DCFLs, not CFLs.
>
> You’re absolutely correct, we will fix it.
>
> >183: The definition of DPDA is missing constraints on $\delta$ that make it deterministic. I think you need to allow $\varepsilon$  as the popped symbol too.
>
> Thanks for catching that, you are again correct. We will revise accordingly.
>
> >088: Do you mean that the number of errors per trace is O(1) with respect to length, not finite?
>
> This is correct, thanks for catching it! We will update it.
>
> >Fact 2.2: In this example, what is the alphabet ?
>
> You can think of the alphabet $\Sigma =\{0,1\}$ and every natural number is encoded in its binary representation.
>
> >Theorem 3.1: Do we assume all the TMs are deciders?
>
> Thanks for bringing this up, this is an excellent point. We do not need the TMs to be deciders, since the language identification task only requires one to correctly identify the set of positive examples (in our case, it is the set of accepted strings of TM). Even if the TM is not a decider, we can identify the positive examples. We cannot distinguish strings that are rejected and strings for which the TM never halts, but this is not required in our model. That is to say, if one wants to learn the TM and correctly identify the set of strings that are accepted, rejected and never halts, then we need TM to be decider. Notice that since the input stream contains positive examples only, the TM must have halted on all these examples.
>
> >310: Does this work if the alphabet is not fixed to {0, 1} ahead of time?
>
> This is a good and nuanced point. Indeed, we can get the same result even if the alphabet isn’t known ahead of time: the learner can figure out the alphabet by observing the accepted strings. As long as its size is finite, this should be possible.
>
> $U$ is the set of states that are ever observed by the algorithm. However, there might be states that never appear. For example, if the adversary always removes the state $s$ in the computational trace, then the algorithm would never see $s$ and $m(x)$ is not 0 whenever $s$ appears in the actual computational trace $c_{M^{\star}}(x)$.
>
> We also allow different instances of the traces for the same $x$ to be edited in different ways.
>
> >388: This statement doesn't make sense to me. Do you mean "not every accepting state is from the set U?
>
> We are sorry for the typo, we want to say “every accepting state is not **far** from the set $U$”. We will revise the text.
>
> >Def 6: What about non-scanning transitions?
>
> Could you please clarify what you mean by non-scanning transitions?

---

### Official Review · Reviewer_evx1 · 2025-10-30

**Soundness:** 4
**Presentation:** 3
**Contribution:** 3
**Rating:** 8
**Confidence:** 2

**Summary:**

An existing result shows that only a very small set of languages can be identified in the limit (i.e., eventually the identifier produces the same prediction forever) from exclusively positive examples provided. The work in this paper demonstrates that a larger class, all Turing Complete languages, can be recognized in the limit when provided with an exact computational trace (the state of one machine which recognizes the language at every step of its computation). Additionally, it demonstrates that this is possible even if there are errors in the trace, with different bounds on the error rates depending on the class of languages discovered.

**Strengths:**

While I did not precisely check every single aspect of the theorems, what I did check appears to be completely accurate, and the theorems are quite interesting in their results.

The fact that the computational trace enables a lot more identifiability is interesting

The text is written quite clearly

**Weaknesses:**

This is primarily a framing issue, but I don't really see the relationship between this work and chain of thought in particular, it seems to be mostly about utilizing information about intermediate states in computational models in order to theoretically learn languages in an unbounded computational setting (with no limits on time to process each sample or the number of samples). In practice, chain of thought uses a very small number of examples and an extremely bounded computation.

Minor errors/suggestions:

You should emphasize early on that “constant number of errors" means a constant per trace, not a constant overall.

Revisiting the regular language example with traces might be helpful, it made the utility of the trace more obviously useful when I went through the example with a trace and realized that the main thing it provides is a distinction between a model that accepts everything (single accept state) and a model that accepts N things (many states).

**Questions:**

The algorithm for identifying robustly only seems to me to use the traces solely to identify the number of states. Is this accurate? If so, it should be explicitly stated in the text, as this sounds like a much weaker assumption than having access to the full traces with errors. If not, the additional information gained should be discussed.

---

> ### Author Response · Authors · 2025-11-21
>
> We thank the reviewer for their time and for offering this useful feedback. We are encouraged that the reviewer found our theorems and results interesting. We address the specific comments below and will revise our manuscript to incorporate these discussions.
>
> >This is primarily a framing issue … bounded computation.
>
> We acknowledge that there is some distinction between our theoretical setting (which is grounded on identification in the limit) and the practical, computationally bounded setting of LLMs. Our motivation is to provide a theoretical framework, inspired by the empirical success of CoT, that is able to illustrate a benefit of having this type of information. In practical LLM training, CoT is widely viewed as revealing the intermediate reasoning steps that lead to the final output (e.g., the steps that a human took to solve a complicated puzzle). Our definition of a computational trace is inspired by this idea by providing the learner with the sequence of internal states and actions taken by a machine to accept an input string; thus this trace can be thought of as the “reasoning-path” of a machine that accepted the given string. We agree that this connection can be made clearer early in the paper. We will revise the paper accordingly.
>
> >You should emphasize early on that “constant number of errors" means a constant per trace, not a constant overall.
>
> Thanks for the suggestion, we will do that in the next version of our manuscript.
>
> >Revisiting the regular language example with traces might be helpful, it made the utility of the trace more obviously useful when I went through the example with a trace and realized that the main thing it provides is a distinction between a model that accepts everything (single accept state) and a model that accepts N things (many states).
>
> Your interpretation is perfectly correct. We also agree it is useful for the reader to explicitly include it in our paper. We will revise it accordingly.
>
> >The algorithm for identifying robustly only seems to me to use the traces solely to identify the number of states. Is this accurate? If so, it should be explicitly stated in the text, as this sounds like a much weaker assumption than having access to the full traces with errors. If not, the additional information gained should be discussed.
>
> This is an excellent point. Indeed, the main technical work in the proofs in the noisy setting is to show that the noisy traces are used to derive a bound on the number of states. We will highlight that in the next version of our work.

---

> > ### Comment · Reviewer_evx1 · 2025-11-22
> >
> > Thank you for engaging with my comments, and for agreeing to make relevant improvements.

---

### Official Review · Reviewer_6Cej · 2025-10-31

**Soundness:** 3
**Presentation:** 2
**Contribution:** 3
**Rating:** 4
**Confidence:** 3

**Summary:**

This paper studies the benefit of CoT from the perspective of language identification in the limit with computation. While Gold proved the impossibility of recognizing most interesting language classes without computational traces, the authors show that as long as each $L\in\mathcal{L}$ is recognizable by some $M\in\mathcal{M}$, the class $\mathcal{L}$ is identifiable by $\mathcal{M}$ in the limit if computational traces are available. Furthermore, the authors consider robust language identification, concluding that identification is achievable with finite error, but robust language identification remains impossible even under diminishing error.

**Strengths:**

1. This work offers an interesting TCS perspective on the role of CoT, connecting the LLM phenomenon with the theory of language identification in the limit.

2. The results on robust identification provide valuable insights into the significance of CoT quality.

**Weaknesses:**

1. The paper lacks intuitive explanations regarding how CoT contributes to identification.

2. There appear to be conceptual gaps between the theoretical model and realistic CoT. For example, the paper’s model consider enumeration over a language, whereas real-world CoT operates on individual instances (i.e., strings $x\in L$). Moreover, while the robust identification results highlight sensitivity to noise, empirical studies suggest that other factors, such as CoT format or length, may outweigh the correctness of CoT.

**Questions:**

See Weaknesses

---

> ### Author Response · Authors · 2025-11-21
>
> We thank the reviewer for their constructive feedback and for recognizing our work as an interesting TCS perspective connecting LLM capabilities with language identification theory. We are encouraged that you found the robust identification results to provide valuable insights. We address your specific comments below. We will revise our manuscript to incorporate part of this discussion.
>
> >The paper lacks intuitive explanations regarding how CoT contributes to identification.
>
> We appreciate this suggestion. Intuitively, the fundamental difficulty in Gold’s model is the "superset problem": since the learner receives only positive examples, it faces a fundamental ambiguity between the target language $K$ and any superset of it. Without negative examples, the learner cannot safely rule out these supersets.
>
> Our model circumvents this impossibility result because the computational trace provides a window into the internal mechanics of the accepting machine, rather than just the set of strings it accepts. For instance, in the case of DFAs, the trace reveals the sequence of states visited. By utilizing these observed transitions, the learner can progressively reconstruct the state transition graph, eliminating the ambiguity inherent in observing strings alone. This intuition extends to Turing Machines. Specifically, our algorithms maintain a subset invariant: they construct machines that accept a (weakly) subset of the target language. As the learner processes more traces, this subset monotonically grows until it converges to the exact target. In the noisy setting, while the learner cannot directly observe correct transitions, we show that identification remains possible provided the noise rate is appropriately bounded. We will include a formalized version of this intuition in Section 3.
>
> >There appear to be conceptual gaps between the theoretical model and realistic CoT. For example, the paper’s model consider enumeration over a language, whereas real-world CoT operates on individual instances (i.e., strings ). Moreover, while the robust identification results highlight sensitivity to noise, empirical studies suggest that other factors, such as CoT format or length, may outweigh the correctness of CoT.
>
> We appreciate the opportunity to clarify the connection between our theoretical model and practical LLM training.
>
> *Model Choice and Enumeration*: First, regarding our model choice, Gold's assumption of a countable class of languages is fairly minimal, as it covers all languages accepted by Turing Machines (and thus all languages expressible by LLMs). Second, regarding "enumeration" vs. "instances": in our framework, the enumeration acts as a formal abstraction for the stream of training data (the corpus). Crucially, our learner is an online algorithm: it processes a single instance (string and trace) at a time and immediately updates its hypothesis. It does not wait to observe the entire enumeration. This resembles an LLM updating its parameters on a batch of tokens. Notice that the computational trace operates on a **single string at a time**, and not the entire enumeration of strings. Importantly, the guarantee is that the algorithm converges to the target after processing a *finite* number of these individual instances, not after processing the entire enumeration.
>
> While this abstraction does not perfectly capture every aspect of practical LLM training, it builds a formal framework to isolate the specific benefit of having CoT-style information versus training only on accepted strings. This allows us to prove strong formal claims: while identification in Gold’s original setting is very limited, the addition of traces makes it highly tractable. We agree that extending this framework to study other aspects of CoT like format and length is an exciting direction for future work.
>
> *Robustness*: Finally, regarding CoT quality, we view our trichotomy of noise tolerance as illustrating a qualitatively interesting phenomenon: for simpler tasks (learning automata), the learner is highly robust to errors, whereas for complex tasks (PDAs, TMs), the tolerance for error diminishes dramatically. This suggests that as task complexity increases, trace correctness becomes crucial.
>
> We hope this discussion clarifies your concerns. Please let us know if you have further questions. We would be very happy to explain in more detail our modeling choices.

---

> > ### Comment · Reviewer_6Cej · 2025-11-28
> >
> > I thank the authors for the detailed response, which addresses most of my concerns. I will raise my score to 6.

---

> > > ### Author Response · Authors · 2025-12-01
> > >
> > > Thank you very much for taking the time to read our rebuttal and for increasing your evaluation of our work. We will incorporate this discussion to the next version of our work. We understand that due to the new policy of ICLR you aren't able to update your official score. We would be grateful if you could communicate your updated assessment to the discussions with the AC.

---

### Meta-Review · Area_Chair_V2PM · 2026-01-02

**Summary:**

This paper provides a theoretical framework for language identification in the limit with Chain-of-Thought (CoT) corresponding the computational traces allowing to improve capabilities of LLMs. The learner is provided not only with examples from a target language but also with computational traces from the machine that accepts them.  Authors show that this information allows one to learn Turing machines.

Based on the initial reviews:

Reviewer 6Cej - Strengths: interesting theoretical result on the role of CoT connecting LLM with theory of language identification in the limit, esults on robust identification provide valuable insights into the significance of CoT quality. On the other hand, paper lacks intuitive explanation on the contribution of CoT, there is conceptual gap between the theoretical model and realistic Cot.

evx1 - Strengths: theorems quite interesting in their results, the  fact that the computational trace enables a lot more identifiability is interesting, text written quite clearly. On the other hand, relationship between the work and chain of thought unclear, her 'constant number of errors' unclear.

WbPo - Strengths:  interesting paper with significant, original theoretical results, results are relevant to the use of CoT to train LLMs,  paper  written clearly with good contextualization of prior work. The robustness results are quite interesting. On the other hand, paper would benefit from clarifications asked by the reviewer.

XJpB - Strengths: very interesting and useful problem, it puts a new spin on an old/classic setting providing another possible contributing factor behind how CoT helps improve models, exposition and motivation clear, proofs well described, related work section thorough and useful.  On the other hand, connection to generation in the limit not very justified, how the results translate into practice is unclear.

Overall, all reviewers have provided a positive feedback arguing that the paper provides interesting and significant results revisiting an old framework in the context of LLMs. Weaknesses are relatively minor and are related to clarification and realistic aspect of the work while the paper is mainly theoretical.
This is a good paper, I propose then acceptance.

**Reviewer Concerns:**

For Reviewer 6Cej authors have provided answers to all the concerns, the reviewer acknowledged that authors addressed most of his concerns.

For evx1, authors provided direct answers to all remarks/questions.

For WbPo, authors answered to all points.

For XJpB, authors provide feedback to all points. Reviewer mentioned that he would keep his positive evaluation.

**Reviewer Scores:**

Reviewer 6Cej gave a 4 and indicated that he would raise his score to 6.

evx1 gave a 8 with low confidence, his feedback indicated that he kept his score.

WbPo gave a 8, it is unclear if the reviewer would have changed his score or not, but at least he would keep it.

XJpB gave a 6 and indicated that he maintained his positive evaluation while mentioning that he hoped that the paper is accepted.

---

### Decision · Program_Chairs · 2026-01-26

Accept (Poster)